# A protocol for the co-creation and usability/ acceptability testing of an evidence-based, patient-centred intervention for self-management of urinary incontinence in older men

Olawunmi Olagundoye *, William Gibson, Adrian Wagg

Division of Geriatric Medicine, Department of Medicine, Faculty of Medicine & Dentistry, College of Health Sciences, University of Alberta, Edmonton, AB, Canada

* olagundo@ualberta.ca

**Data Availability Statement:** No datasets were generated or analysed in this protocol. All relevant

## Abstract

Male urinary incontinence (UI) is most prevalent in older men, with one in three men aged 65 and above having problems maintaining continence. Addressing health inequalities, male-female disparities in continence services, and low health-seeking among men emphasizes the necessity for co-creating an intervention that empowers them to self-manage their UI. We aim to co-create a self-management intervention with an older men and Health care provider (HCP) group and assess its usability and/or acceptability among older men with UI. The intervention mapping (IM) framework, a co-creation strategy, will be used to co-create a self-management tool, followed by usability and/or acceptability testing. The study will be guided by the first four IM steps: the logic model of the problem, the logic model of change, program/intervention design, and program/intervention production, followed by preliminary testing. A participatory group of older men with UI recruited from an existing group of patient partners, and continence care experts will be involved in all steps of the IM process. Usability and/or acceptability testing will be conducted on a sample of 20 users recruited through seniors' associations and retirement living facilities. After accessing the self-management tool for a week, participants will complete a product usability testing scale (aka System Usability Scale-SUS) and/or an acceptability test, depending on the preferred mode(s) of intervention delivery. Data will be analyzed using descriptive statistics. A benchmark overall mean usability score of 70 represents a good/usable product, based on the large database of SUS scores.

## Introduction

### Impact of UI on older men

UI is defined by the International Continence Society as the complaint of any involuntary leakage of urine [1]. In addition to the adverse effects on physical health, UI is associated with

data from the study will be made available upon study completion.

**Funding:** The proposed study will be funded by the UHF (Muhlenfeld Family Foundation) through a private donation held by my PhD supervisor, Dr. Adrian Wagg. No formal award letter was issued for this private donation. The funders had no role in study design, data collection and analysis, decision to publish, or preparation of the manuscript.

**Competing interests:** The authors have declared that no competing interests exist.

feelings of shame and stigma. It fosters dependence and heightens care demands on family members or friends of an older individual, significantly affecting their wellbeing. UI leads to increased healthcare expenses for older adults [2, 3]. It is also often seen as a sign of incompetence in older adults, which can greatly harm their self-esteem [4].

Men are often reluctant to address their incontinence with healthcare providers. Available data indicate that only one out of five men with symptoms seek medical care. According to an epidemiological survey conducted in the U.S. that involved a racially and ethnically diverse sample of men and women across different socio-economic classes, men sought care half as often as women, with only 22% seeking assistance compared to 45% of women [5–7]. Older men who hold significant roles in traditional family structures are especially vulnerable to feeling incompetent [8]. Men often hold key positions in various contexts, such as heads of households in patriarchal family structures, leaders in religious and community settings, political figures, and key decision-makers in workplaces and agricultural communities [9, 10]. The positions they hold highlight their social influence and the impact that vulnerabilities like urinary incontinence can have on their sense of competence.

UI predicts institutionalization and also early mortality among older males living in the community and in nursing homes [11]. Despite the comparable impact of UI on quality of life for both men and women, male UI has been given less focus in research and discourse [12]. Numerous studies have been conducted on the prevalence, diagnosis, and treatment of urinary incontinence in older women as opposed to men [13–16].

The likelihood of experiencing psychological distress increases when a condition perceived as stigmatized is combined with reduced independence, social isolation, and insufficient professional assistance or support [8]. Men are 90% more likely to restrict their outings because of incontinence, which increases their risk of social isolation, itself a risk for cognitive decline and physical deconditioning [17–19].

## Urinary incontinence interventions

In order to effectively manage UI, particularly in older men, health professionals should integrate psychological, behavioural, and sociocultural interventions [8]. In a survey in the north of England, older people delayed seeking help because some believed UI was a normal part of aging, some were embarrassed and others were unaware that help was available [20]. Therefore, it is important to provide older adults experiencing UI with education and counseling aimed at helping them better understand the range of treatment options available [8]. Healthcare professionals and caregivers must dispel the commonly held view that UI is normal for older age [8].

Understanding the health choices and treatment preferences of older men to attain outcomes that align with their goals is important if optimal health and quality of well-being outcomes are to be attained. For example, in a study by Hampson and colleagues, treatment attributes that mattered to older men with stress UI included likelihood of dryness, treatment simplicity, surgical avoidance, potential need for future interventions, and understanding others' experiences of treatment regret or satisfaction [21].

UI management in older men should encompass conservative measures, behavioral and physical therapies, as well as drug treatments with the acknowledgement that some men may require surgical intervention for their UI [22].

Conservative management includes: (i) Addressing comorbidities that contribute to lower urinary tract symptoms, including UI. (ii) Treating constipation, which is closely associated with UI. (iii) Utilizing containment devices, particularly for patients who have either not responded to or have declined treatment, or for those for whom treatment may not be suitable

[23]. (iv) Lifestyle modifications, targeting factors such as caffeine intake, obesity, smoking, diet, and physical activity, which are associated with UI [23]. Behavioural and physical therapies, such as bladder training, prompted voiding, and pelvic floor muscle training (PFMT), electrical and magnetic stimulation, and percutaneous tibial nerve stimulation (PTNS), represent additional non-surgical management options [23].

In their investigation into self-care practices among older men and women coping with UI, Johnson and colleagues discovered that over a quarter of respondents employed such practices [17]. These included utilizing containment devices, such as disposable pads (36.8% [271/787]), reducing the number of outdoor visits (27.6% [217/787]), and moderating fluid intake (36.6% [288/787]). Women were more likely than men to utilize disposable pads and engage in exercise therapy [17].

Self-management interventions are designed to improve an individual's medical, behavioral, and emotional condition thereby giving them control over their symptoms [22, 24]. Informing and advising patients about lifestyle modifications is rarely sufficient to improve their health. This limitation has led to the development of self-management programs, which combine information and advice with techniques to promote behavioral change. Self- management programs aim to help patients manage their disease by enhancing their problem-solving and goal-setting skills [22].

Self-management is a proven intervention for chronic diseases It emphasizes patient-centered strategies to address challenges, with a focus on modifiable risk factors [25]. Self- management is a promising strategy for managing chronic conditions such as UI. It requires individuals to be involved in identifying challenges and solving problems related to their health [22].

By engaging stakeholders in intervention mapping, programs and interventions are more likely to be relevant, feasible, and ultimately successful [26].

No existing self-management interventions for UI focus on the needs of older men and none incorporate the perspectives of older men into their development. Furthermore, the health inequalities and disparities in continence services for men, and a low level of health seeking in men with UI make it crucial to incorporate their perspectives into intervention development to ensure optimal outcomes [8, 27].

So far, self-management intervention packages for men have targeted uncomplicated lower urinary tract symptoms (LUTS) generally and mostly in men with prostate disease (Benign Prostatic Obstruction). These packages vary in their components, recommendations, and outcomes [28–30].

Changing patient behaviour and maintaining these changes are challenging tasks. There are many interventions that require patients and facilitators to dedicate time and effort. For example, changing caffeine and alcohol consumption patterns may significantly affect quality of life, potentially making them impracticable [30].

Lifestyle factors that can be modified through behavioural adjustments include managing fluids, avoiding certain drinks (caffeine, carbonated beverages, alcohol), prompted voiding (timely reminders to void in those with cognitive impairment), bladder training (a systematic approach to modifying voiding patterns), and pelvic floor muscle exercises/training (PFME/PFMT) [31].

A Cochrane meta-analysis of UI in women suggests that PFMT could serve as a viable first-line therapy for UI, effectively curing or improving symptoms of SUI and other types of UI [32].

In Wu et al.'s meta-analysis of male UI after radical prostatectomy, PFME guided by a therapist (G-PFME) aided urinary continence recovery [33]. Similarly, a Cochrane review meta-analysis demonstrated that PFME could accelerate recovery, particularly during the third to sixth months postoperatively, but is unlikely to be effective beyond 12 months [34]. In addition, evidence, primarily from studies focused on women, indicates that enhancing pelvic floor

function through PFME could help suppress bladder contractions among individuals with Overactive Bladder (OAB) [32, 35].

Both prompted voiding (PV) and bladder training (BT) have demonstrated efficacy in both men and women [31]. BT may assist physically and cognitively functional individuals, though symptom improvement may take months. In contrast, PT can enhance continence in both functional and non-functional individuals but demands extensive and consistent staff management techniques [23].

It is essential to provide older men with education about aids and appliances for managing urinary incontinence so they can feel more confident and comfortable. These tools, ranging from absorbent pads to urinary catheters to penile clamps, offer practical solutions tailored to individual needs [23]. By discussing these options and providing resources such as www.continenceproductadvisor.org, men can explore a wide range of products that align with their preferences. Providing support in finding the right product ensures they can effectively manage their condition while maintaining dignity and independence.

In their work, Brown *et al* undertook a formal consensus process in defining lifestyle modifications that were likely to be effective in the self-management of uncomplicated LUTS and incorporated these recommendations into a self-management intervention [30]. However, this work neglected the engagement of men with LUTS.

Yap *et al.*'s randomized controlled trial incorporating a self-management program and standard care resulted in a statistically significant improvement in lower urinary symptoms (LUTS) in the group receiving both interventions, compared to those receiving standard care only [36], demonstrating the value of a self-management program. Brown et al reported a decrease in the proportion of patients who needed medication or surgery over 12 months and an improvement in self-rated LUTS in the group of patients who used the self-management program [29].

**Modes of intervention delivery.**   A randomized trial of a continence promotion workshop, paired with an evidence-based self-management booklet, successfully alleviated UI symptoms among community-dwelling older women [16, 37].

An online personalized self-management platform, developed by a group of researchers and subsequently tested for utility among experts and usability among men with uncomplicated LUTS, yielded fair results [38]. To improve on this, and taking into account the levels of health literacy and digital literacy of older men [39, 40], we will employ a community-based participatory co-creation approach, actively involving stakeholders throughout the development process [41].

This study will build on systematically synthesized evidence on risk factors for UI in older men [27, 42], as well as continence experts' consensus on factors that are potentially modifiable, and older men's perspectives on what they find practicable and are willing to modify, as well as their preferred mode of intervention delivery to co-create a self-management intervention.

## Objectives

i.  To co-create a self-management intervention with an older men and HCP group

ii.  To determine its usability and/or acceptability among older men with UI.

## Materials and methods

As part of a larger study designed to define, prioritize and co-create a self-management intervention for older men, this study leverages the results of preceding sub-studies, including the

first four steps of the systematic stepwise method of the intervention mapping framework to co-create a self-management tool(s), followed by usability/acceptability testing.

Intervention mapping is a systematic framework used in health promotion and public health to develop evidence-based interventions. It involves a structured process that integrates theory, empirical evidence, and input from stakeholders to create effective health programs. This approach ensures that interventions are well-grounded in theory, tailored to the needs of the target population, and capable of achieving measurable health outcomes [41]. Globally, IM is widely used for behavioural change interventions. There is a significant increase in disease prevention behaviours associated with IM-based interventions compared with placebo controls [43].

The steps guiding this approach are detailed below:

Step 1 –The logic model of the problem: This model helps us understand specific health issues by identifying their causes, consequences, and available resources to address them [41]. Clarifying these elements guides us in selecting effective strategies and actions to address them.

According to our needs assessment, self-management is an unmet need identified through preliminary discussions with older men with UI attending the continence clinic as well as men on our patient advisory board. A review of the literature on male UI, components and outcomes of self-management of UI, a scoping review of risk factors for UI in older men, a sequential multimethod consensus study and older men's survey, and the formation of an older men's and experts' participatory group are also covered in step 1.

*Formation of a participatory group of older men with UI and continence care experts*: Consenting older men with UI will be recruited from an existing group of patient partners. Multidisciplinary experts in continence care will be recruited via the Canadian Continence Foundation (CCF). They will be involved in all steps of the IM process.

The group will consist of a minimum of six stakeholders, with 50% being older men with UI and 50% being continence care experts. Experts will be recruited through snowballing and formal calls through the CCF, and handbills and posters will be distributed to patient partners/patient advisory boards, ethnic associations, and senior organizations.

Step 2 –The logic model of change: Specification of performance objectives, determinants, and change objectives. This model describes how a health program contributes to individuals' wellbeing, describing the activities to be carried out and the immediate benefits to be expected. It also anticipates long-term outcomes like improved health behaviours or conditions. Following this plan ensures the program is well-structured and allows measurement of its success in achieving the desired outcomes [41].

In Step 2, the group will identify and state expected behavioural and environmental outcomes, identify performance objectives for these outcomes, select determinants for these outcomes, develop matrices of change objectives, and construct a logic model of change. For this step, data from the literature review, empirical findings, older men's and experts' input are triangulated to inform a logic model of change.

Step 3 –Program/intervention design: Program design (selection of intervention strategy that delivers the method in a way that fits the needs of the priority group and the program setting). This step involves generating the program scope and sequence, choosing theory and evidence-based methods, and creating practical applications to deliver change methods.

The group will initially brainstorm ideas for the program/intervention and select theory- and evidence-based behavior change methods based on the determinants requiring modification. During this stage, program objectives will be organized or categorized according to these determinants.

Theoretical methods conducive to achieving the program objectives will be identified and then translated into practical applications or strategies. A theory-based change method refers

to a technique for altering a behavioral determinant of an individual or environmental agent, whereas a practical application is a specific strategy that implements the method in a manner suitable for the priority group and program setting. Certain methods may address multiple determinants, while others may be tailored to specific ones.

Step 4 –Program/intervention production: Program creation and preliminary testing. The various applications/strategies selected in Step 3 will be organized and produced. The group will decide on the overall structure and vehicles of the program. Following this, we will refine the structure and organization of the selected strategies until they are satisfactory. We will then proceed to plan, draft, refine, and produce materials that are culturally relevant and appealing. These materials will undergo pre-testing to ensure their functionality. Pre-testing involves either in-house experts' alpha testing for digital interventions or utility testing for educational programs. Any issues identified during pre-testing guide program/intervention revisions.

In the proposed study, step 4 will be followed by usability/acceptability testing among a sample of older men/potential end users.

Step 5 (Program/intervention implementation plan) and Step 6 (Evaluation plan) will form part of the next study in the overarching self-management program study.

## Usability/Acceptability testing

Sampling and recruitment strategy: A usability and/or acceptability survey (S1 Appendix. Self-management intervention usability and acceptability survey) will be conducted among consenting older men recruited from the community, including seniors' associations and retirement living facilities.

For effective usability testing, a sample size of at least 20 users is recommended, sufficient to detect 98% of product problems [44].

## Data collection procedure and instruments

For an app, participants will receive access to the self-management tool for a week, after which they will complete the product usability testing scale (aka System Usability Scale-SUS) as well as acceptability testing. For a potential in-person intervention, such as an educational workshop, participants will complete an acceptability survey at the conclusion of the workshop.

Acceptability will be assessed using the acceptability domain questions from the feasibility studies' guidelines by Pearson *et al.* [45].

## Statistical analysis

Usability and/or acceptability data will be analysed using descriptive statistics. The Statistical Package for Social Sciences for Windows (SPSS, version 26.0; IBM Corp, Armonk, New York) will be used for data analysis. Continuous variables will be presented in means and standard deviation (SD), while categorical variables will be presented in proportions. Based on the large database of SUS scores over several years, a benchmark overall mean usability score of 70 (median 70.5) represents a good/usable product [46].

## Ethical clearance

This study protocol has been approved by the University of Alberta's Health Research and Ethics Board (Ethics ID: Pro00141446).

Administrative permission will be sought from the seniors' associations and retirement living facility managements. Informed consent will be sought from all the participants.

## Significance of the anticipated results

The results will bridge a significant gap in the evidence by providing empirical data and stakeholder inputs in fostering an understanding that will inform production, implementation and maintenance of a patient-centred intervention to improve treatment outcomes and UI-related quality of life. The possibility of deploying the tool as an app increases accessibility for prospective end-users beyond the study setting. It also increases the ease of evaluating process and outcome/effect measures and guarantees continuous generation of individual progress reports for users and aggregated data for research. A mobile app customized to personal UI risk factors and self-management needs/goals with action reminders/push notifications will enhance performance of tasks that will translate into achieving the desired outcomes.

An in-person delivered intervention, such as an educational workshop offers direct interaction, immediate feedback, and personalized guidance, enhancing learning outcomes. They facilitate networking and relationship building among participants, fostering a supportive community and shared experiences. Face-to-face meetings foster mutual understanding and personal growth by creating an environment where individuals can openly share insights and concerns.

## Dissemination of results

Members of the co-creation participatory group will receive a report on the pretesting and usability or acceptability testing. Findings will be disseminated to the public through presentations at conferences or workshops, peer-reviewed publications, institutional research repositories, health blogs, and other social media platforms such as LinkedIn, X (Twitter), and Instagram. Healthcare professionals will receive information through continuing professional development seminars and workshops. Educational materials such as infographics will be distributed to patients through clinics and the public through strategic partners such as seniors' associations.

## Supporting information

**S1 Appendix.**
(PDF)

## Acknowledgments

The research study will contribute towards a PhD degree award for OO.

## Author Contributions

**Conceptualization:** Olawunmi Olagundoye, Adrian Wagg.

**Funding acquisition:** Adrian Wagg.

**Methodology:** Olawunmi Olagundoye, William Gibson, Adrian Wagg.

**Project administration:** Olawunmi Olagundoye.

**Resources:** William Gibson, Adrian Wagg.

**Supervision:** Adrian Wagg.

**Writing – original draft:** Olawunmi Olagundoye.

**Writing – review & editing:** Olawunmi Olagundoye, William Gibson, Adrian Wagg.

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
