## [Decision Letter · Decision Letter 0]

2 Jul 2024

PONE-D-24-22945A protocol for the co-creation and usability/acceptability testing of an evidence-based, patient-centred intervention for self-management of urinary incontinence in older menPLOS ONE

Dear Dr. Olagundoye,

Thank you for submitting your manuscript to PLOS ONE. After careful consideration, we feel that it has merit but does not fully meet PLOS ONE’s publication criteria as it currently stands. Therefore, we invite you to submit a revised version of the manuscript that addresses the points raised during the review process.

We look forward to receiving your revised manuscript.

Kind regards,

Alison Parker

Academic Editor

PLOS ONE

Journal Requirements:

"The proposed study will be funded by the UHF (Muhlenfeld Family Foundation) through a private donation held by Dr. Adrian Wagg. 

No formal award letter was issued for this private donation."

Reviewers' comments:

Reviewer's Responses to Questions

**Comments to the Author**

1. Does the manuscript provide a valid rationale for the proposed study, with clearly identified and justified research questions?

Reviewer #1: Yes

Reviewer #2: Yes

2. Is the protocol technically sound and planned in a manner that will lead to a meaningful outcome and allow testing the stated hypotheses?

Reviewer #1: Yes

Reviewer #2: Yes

3. Is the methodology feasible and described in sufficient detail to allow the work to be replicable?

Reviewer #1: Yes

Reviewer #2: Yes

4. Have the authors described where all data underlying the findings will be made available when the study is complete?

Reviewer #1: Yes

Reviewer #2: Yes

5. Is the manuscript presented in an intelligible fashion and written in standard English?

Reviewer #1: Yes

Reviewer #2: Yes

6. Review Comments to the Author

You may also provide optional suggestions and comments to authors that they might find helpful in planning their study.

Reviewer #1: The protocol provides the grounding for what looks to be a robust and interesting study, with a clear methodology. I just have a couple of minor suggestions which may strengthen it further

The introduction refers to only 22% of men seeking assistance- is this just in high income countries or do we know if this is a global figure that also takes into account low- and middle income countries? It would be good to clarify this. Similarly in the same paragraph, it may be useful to name some examples of contexts in which men hold significant societal (even traditional) roles.

In the section on urinary incontinence interventions on page 11, there is a line stating how older people perceived UI as a part of aging. Is this from a particular research study? Better context for this study needs to be added- which older people? Where?

On page 13 I had a couple of minor comments. The authors could add 1-2 examples of the stakeholders they plan to engage in the intervention mapping etc. The paper rightly highlights the lack of focus on older men. A couple of references and examples could be added to highlight the existing significant focus on older women, just for context and to further emphasise this significant gap. In the same paragraph, should the paper be referring to 'health seeking behaviours' rather than just health seeking?

Finally on page 17. How will recruitment take into account diversity, are there any specific efforts being made to include men from ethnic minorities/First Nations or indigenous groups?

This is a clear protocol for a promising study.

Reviewer #2: This is a well-written manuscript focusing on a sadly under-addressed area of key importance in public health, incontinence in older males.

The manuscript describes a study that will be undertaken to gain understanding in how self-management for UI in elderly males can be developed.

The literature review demonstrates a deep understanding of the subject area and relates to well posed aim and objective. The methods are appropriate and show a keen appreciation of how clinical research, particularly in stigmatised subjects such as incontinence, should be conducted.

Very minor ammendments are recommended, as outlined below - which do not warrant re-review. Hence a recommendation of accept.

Excellent and commendable research.

Abstract:

Usability/acceptability testing - please rephrase to avoid use of / in the text.

Can you briefly define 'logic model' as part of the intervention mapping (IM) framework - to make the abstract more accessible to a wider audience? E.g. the logic-model (which defines the ...)

In the main body - please re(define) IM, (not just include in the abstract) and provide a short defintion to make the paper self-contained.

7. PLOS authors have the option to publish the peer review history of their article (what does this mean?). If published, this will include your full peer review and any attached files.

Reviewer #1: **Yes: **Amita Bhakta

Reviewer #2: No

---

## [Author Response · Author response to Decision Letter 0]

26 Jul 2024

Dear Reviewers,

We appreciate your comments and the opportunity to improve our manuscript. A response letter uploaded alongside the revised manuscript shows our response and the pages where changes have been made to the manuscript.

Sincerely,

Olawunmi Olagundoye (On behalf of the authors)

---

## [Editor Report · Decision Letter 1]

30 Jul 2024

A protocol for the co-creation and usability/acceptability testing of an evidence-based, patient-centred intervention for self-management of urinary incontinence in older men

PONE-D-24-22945R1

Dear Dr. Olagundoye,

We’re pleased to inform you that your manuscript has been judged scientifically suitable for publication and will be formally accepted for publication once it meets all outstanding technical requirements.

Kind regards,

Alison Parker

Academic Editor

PLOS ONE
---

## [Editor Report · Acceptance letter]

2 Aug 2024

PONE-D-24-22945R1 

PLOS ONE

Dear Dr. Olagundoye, 

I'm pleased to inform you that your manuscript has been deemed suitable for publication in PLOS ONE. Congratulations! Your manuscript is now being handed over to our production team.

Kind regards, 

on behalf of

Dr. Alison Parker 

Academic Editor

PLOS ONE